# Comprehensive analysis of prostate cancer life expectancy, loss of life expectancy, and healthcare expenditures: Taiwan national cohort study spanning 2008 to 2019

**Pin-Chun Liu**[1,‡], **Yi-Sheng Lin**[1,‡], **Yen-Chuan Ou**[1,*], **Chao-Yu Hsu**[1,‡], **Min-Che Tung**[1,‡], **Ying-Ming Chiu**[2,3,4,‡]

1 Division of Urology, Department of Surgery, Tungs' Taichung Metroharbor Hospital, Taichung City, Taiwan, 2 Departments of Allergy, Immunology, and Rheumatology, Tungs' Taichung MetroHarbor Hospital, Taichung City, Taiwan, 3 Departments of big data research center, Tungs' Taichung MetroHarbor Hospital, Taichung City, Taiwan, 4 Departments of Post-Baccalaureate Medicine, College of Medicine, National Chung Hsing University, Taichung City, Taiwan

‡ These authors contributed equally to this work.
* ycou228@gmail.com

## Abstract

Prostate cancer (PCa) is the second most commonly diagnosed cancer worldwide and the 5th leading cause of death from cancer for men in Taiwan. The incidence of synchronous metastatic PCa in Taiwan is higher than U.S. and Europe. We aim to present the latest life expectancy (LE), loss of LE, and lifetime cost associated with PCa in Taiwan. The PCa data are based on Taiwan Cancer Registry and National Health Insurance Database. Total 30,207 new cases of PCa were recorded during 2008-2019 nationwide. LE, estimated loss of LE and lifetime cost were stratified by age, cancer stage, Gleason score, grade group and serum PSA level at diagnosis. We compared LE and healthcare cost outcomes between synchronous metastatic PCa patients in 3 age groups. Among the 30,207 new cases, the low to intermediate risk groups, high-risk groups, and regional and metastatic PCa accounted for 54.1%, 13.2%, and 32.6% of cases, respectively. A considerable proportion of synchronous metastatic PCa was noted in Taiwan when compared with the U.S. For synchronous metastatic PCa, the highest LE is 9.22 years for ages 20-64 years, followed by ages 65-74 (8.29 years) and ages 75-89 years (4.58 years). The loss of LE in the three age groups is 13.63, 6.75, and 3.87 years, respectively. The healthcare cost of synchronous metastatic PCa in all age groups is higher than the average cost for PCa patients in Taiwan. This study provides real-world evidence to support health care policy-making and clinical decisions regarding PCa. Due to the high proportion of synchronous metastatic PCa in Taiwan, the findings of this analysis emphasize the importance of early detection of PCa, which can save LE and decrease the total cost burden on the healthcare system.

**Data availability statement:** All relevant data are available from the Taiwan Cancer Registry database. Due to privacy and data security regulations, the data cannot be shared publicly. Researchers can request access by applying for data use permission and undergoing a review by the relevant units. Detailed application procedures can be found on the Agency for Public Health Research website: https://www.apre.mohw.gov.tw/.

**Funding:** The author(s) received no specific funding for this work.

**Competing interests:** The authors have declared that no competing interests exist.

## Introductions

Prostate cancer (PCa) was the cancer with the highest incidence for men in 114 countries and the leading cause of cancer-related deaths among men in 56 countries in 2017 [1]. It accounted for 191,930 new cases and 33,330 deaths among U.S. males in 2020. From 2012 to 2017, the incidence of PCa in the United States was 104.1 per 100,000 people, with a mortality rate of 19.1 per 100,000 people [2]. In Taiwan, PCa was responsible for the fifth cancer cause of death among male based on the statistics of cause of death in 2019 announced by the ministry of health and warfare. The estimated crude death rate caused by PCa is 13.1 per 100,000 persons [3]. Statistics indicate a progressive increase in the incidence rate of PCa in Taiwan. Overall, the age-standardized incidence of PCa between 2008 and 2019 increased by 11.22% (from 24.61 to 35.83 per 100,000 people). The accurate reason could be complicated and multifactorial; proposed explanations include an aging population, the popularization of blood tests, westernized diets, and lifestyle alterations. A previous study revealed a wide variation in the incidence and mortality rates of PCa in Asians. Compared with other Asian countries, the incidence of PCa in Taiwan is second only to Australia, New Zealand, and Turkey [4].

The treatment of PCa imposed a great impact on both individuals and the country's financial burden. Gustavsen, Gary et al. indicated that the cumulative cost of managing localized PCa on a per-patient was estimated to be USD $46,193 over 5 years and USD $110,993 over 10 years [5]. Smith-Palmer, J et al. mentioned that total overall cost in the first year after diagnosis of PCa was EUR 385 million in France, followed by EUR 244 million in Germany and then the United Kingdom at EUR 117 million per year in 2006 [6]. PCa also causes a great number of deaths worldwide annually. A study by Withrow, Diana et al. mentioned that 3.5 million person-years of life were lost due to PCa in males over 50, with 40% of years of life lost occurring in those aged over 75 [7].

However, most of the previous studies are based on materials and documents from Western countries. Hwang et al. developed a novel semiparametric method to improve the accuracy of estimates of life expectancy (LE) and loss of life expectancy (loss of LE) from the date of diagnosis [8,9]. In this study, we estimated the PCa patients LE, loss of LE and lifetime health care expenditures, stratified by age and stage in the Taiwanese population with up to 12 years of follow-up.

## Materials and methods

We identified a study cohort by using the database of Taiwan Cancer Registry (TCR) database from 2011 to 2017, and patients with PCa were identified by ICD-O-3 topography code: C61.9. Patients who were younger than 20 years of age or older than 90 years of age were excluded. TCR provides complete and high quality information on cancer medical records in Taiwan [10]. The data in the database has been de-identified, making it impossible to identify the subjects. Our study used data from TCR including gender, date of diagnosis, age at diagnosis, PCa stage (stage grouped according to AJCC cancer staging 8th edition), prostatic specific antigen (PSA), and Gleason's score. PSA value was divided into five levels: < 4, 4-9, 10-19, 20-50 and > 50 ng/ml. Gleason's score also was divided into five levels, ≤ 6, 3 + 4, 4 + 3, 8 and 9-10. Gleason's score 3 + 4 indicates the primary pattern was 3 and the secondary pattern was 4. In contrast, 4 + 3 means the primary pattern was 4 and the secondary pattern was 3.

The survival status of patients was verified by linkage with the National Mortality Registry. All patients were followed until the end date of follow-up, the date of death or December 31, 2019, whichever came first. Survival function was generated via the Kaplan-Meier method until the end of follow-up. LE was estimated by a semiparametric survival extrapolation method, which was proposed by Hwang and Wang [8,9]. This method has been

applied to various diseases such as rheumatoid Arthritis [11], dialysis [12,13], schizophrenia [14] and cancer [15]. Based on the life table of the National Vital Statistics of Taiwan, a reference group matching the age, sex and calendar year of PCa patients was generated by the Monte Carlo method. Assuming that the survival rate of the reference group was better than patients, the survival ratio between the patients and the reference group was logit-transformed each month. The survival rate of patients for the next month could be predicted by fitting restricted cubic spline models. Further extrapolation of survival rate was predicted by using a rolling extrapolation algorithm. Assuming the survival prediction for the next month was true and ignoring the data of the first month used in the previous prediction, do the prediction again until the survival rate approaches zero. The area under the survival curve of patients was defined as LE. Besides, the difference of LE between the patients and the reference group represents loss of LE. Loss of LE could be used as a measure of life loss because of disease.

The health care expenditures of each patient were confirmed by linkage with the national health insurance reimbursement database. The monthly average cost was calculated by adding the monthly costs of all patients, including the costs of inpatient and outpatient, and dividing by the number of these patients who were still alive in each month. Assuming that the patient's health care expenditures increase in the months before death, the average cost function is estimated by weighting the average cost of the patient's months before death. Lifetime cost can be calculated via the monthly average cost function multiplied by the monthly survival function. Annual national health insurance expenditures were adjusted based on the 2019 Consumer Price Index and the exchange rate (1 USD = 30.93 TWDs), and then the estimated costs were adjusted at the annual discount rate (3%). The estimates of LE and lifetime costs were calculated by the free R package iSQoL2, and the study was reviewed and approved by the institutional review board of Tungs' Taichung MetroHarbor Hospital (IRB number:110034). The need for participant informed consent was waived by the ethics committee.

## Results

A total of 30,207 patients with PCa were included in the study. Table 1 summarized the estimated LE, loss of LE, lifetime cost and mean cost per year stratified by age and cancer stage. Survival curves of stages 1 to 3 were similar or even better than the matched reference group, which violated the assumption of LE estimation method that the survival rate of the patients should be worse than the reference group. Therefore, the LE and lifetime cost of those groups could not be estimated. In stage 4, the loss of LE for the 20-64 age groups was the highest at 13.63 years, while the loss of LE for the 75-89 age groups was the lowest at 3.87 years. (Fig 1) The younger the patients, the greater the LE and loss of LE. For the lifetime cost, the 20-64 age group is the highest, with 84,089 US dollars, while the 75-89 age group is the lowest, with 35,841 US dollars. The results were similar in the mean cost per year, with $9,893 for the 20-64 age group and $4,573 for the 75-89 age groups.

In Table 2, the patients were stratified by age and PSA value. The survival rate of level 4-9 and 10-19 ng/ml were similar or even better than the reference group so LE can't be estimated, either. However, the group of < 4 ng/ml has worse survival than the reference group. We reviewed the medical history of those patients and the result showed that a higher proportion of transurethral resection of prostate (TURP) was performed one year before cancer diagnosis in this group (S1 Fig). The PSA value usually decreased after TURP [16], so we think the PSA level can't represent the actual value in < 4 level group. In other groups of 20-50 and > 50 ng/ml, regardless of age group, the higher the PSA value, the lower of LE, the greater loss of LE, and higher mean cost per year (Fig 2)

**Table 1. Estimation of life expectancy, loss of life expectancy, lifetimes cost (USD) and means cost per year for prostate cancer, stratified by age and stage.**

| Age group | Stage | N | No. of death | Age* | LE (SE)[#] | Loss of LE (SE) | Lifetime cost (SE) | Cost per year[$] |
|---|---|---|---|---|---|---|---|---|
| 20–64 | | | | | | | | |
| | Stage 1 | 1,016 | 39 (3.84%) | 60.5 ± 4.03 | – | – | – | – |
| | Stage 2 | 2,733 | 137 (5.01%) | 60.34 ± 3.97 | – | – | – | – |
| | Stage 3 | 718 | 53 (7.38%) | 60.54 ± 3.91 | – | – | – | – |
| | Stage 4 | 1,652 | 732 (44.31%) | 59.95 ± 4.2 | 9.22 (1.36) | 13.63 (1.36) | 84,089 (5,957) | 9,893 |
| 65–74 | | | | | | | | |
| | Stage 1 | 1,413 | 139 (9.84%) | 69.92 ± 2.83 | – | – | – | – |
| | Stage 2 | 5,057 | 631 (12.48%) | 70.2 ± 2.9 | – | – | – | – |
| | Stage 3 | 1,594 | 210 (13.17%) | 70.42 ± 2.79 | – | – | – | – |
| | Stage 4 | 3,173 | 1,445 (45.54%) | 70.44 ± 2.9 | 8.29 (0.48) | 6.75 (0.48) | 68,109 (2,825) | 8,690 |
| 75–89 | | | | | | | | |
| | Stage 1 | 1,053 | 335 (31.81%) | 80.38 ± 3.88 | – | – | – | – |
| | Stage 2 | 5,093 | 1,742 (34.20%) | 80.57 ± 3.81 | – | – | – | – |
| | Stage 3 | 1,681 | 606 (36.05%) | 80.51 ± 3.79 | – | – | – | – |
| | Stage 4 | 5,024 | 3,268 (6505%) | 81.48 ± 3.96 | 4.58 (0.14) | 3.87 (0.14) | 35,841 (918) | 4,573 |

*Age presented as mean±standard deviation;

[#]SE, standard error of the mean;

[$]1 dollar = $30.93 new Taiwan dollar; LE, Life expectancy, in year; loss of LE, loss of life expectancy.

Table 3 and Fig 3 outlined the estimated LE, loss of LE, lifetime cost and mean cost per year stratified by age and Gleason's Score. The groups of Gleason's Score ≤6 and 3+4 could not estimate their LE and lifetime cost also because of similar or even better survival. Regardless of age group, the group of Gleason's Score 9-10 had the greatest loss of LE. The highest lifetime cost was the group of Gleason's Score 8. However, the highest mean cost was the group of Gleason's Score 9-10.

## Discussion

This study estimated the LE, loss of LE, and lifetime health care costs of PCa. LE was estimated by a semiparametric survival extrapolation method, which was proposed by Hwang and Wang [9,10]. This method has been applied to various diseases such as rheumatoid arthritis [11], dialysis [12,13], schizophrenia [14] and cancer. It is the first time it has been used to analyze the LE of PCa. The LE (shown in Table 1) of stages 1 to 3 PCa were similar or even better than the matched reference group. The same result can be observed when LE, loss of LE and lifetime healthcare costs has been stratified by age and PSA or age and Gleason's score (shown in Tables 2, 3). There are some possible explanations. First, the early stage PCa has a good prognosis under the care of Taiwan's national health system. The majority of the treatment can be subsidized by the national health system. Early-stage PCa does not affect a patient's average LE. There is no significant difference in the statistics of LE between stages 1-3 of PCa, whether in the 20-64 year old group or the 75-89 year old group. Second, being diagnosed with cancer raises awareness of a patient's own health. Patients become more conscious of their health after being diagnosed with PCa. They also come to the hospital more often for follow-up of PCa, and if they have other health problems, they will be found earlier.

Our study also revealed another characteristic of PCa in Taiwan: a high proportion of synchronous metastatic PCa across all three age stratification. Synchronous metastatic PCa accounts for 27% in age group 20-64, 28.2% in age group 65-74, and 39% in age group over 75, respectively. Among all newly diagnosed PCa patients, Taiwan exhibits a considerably high

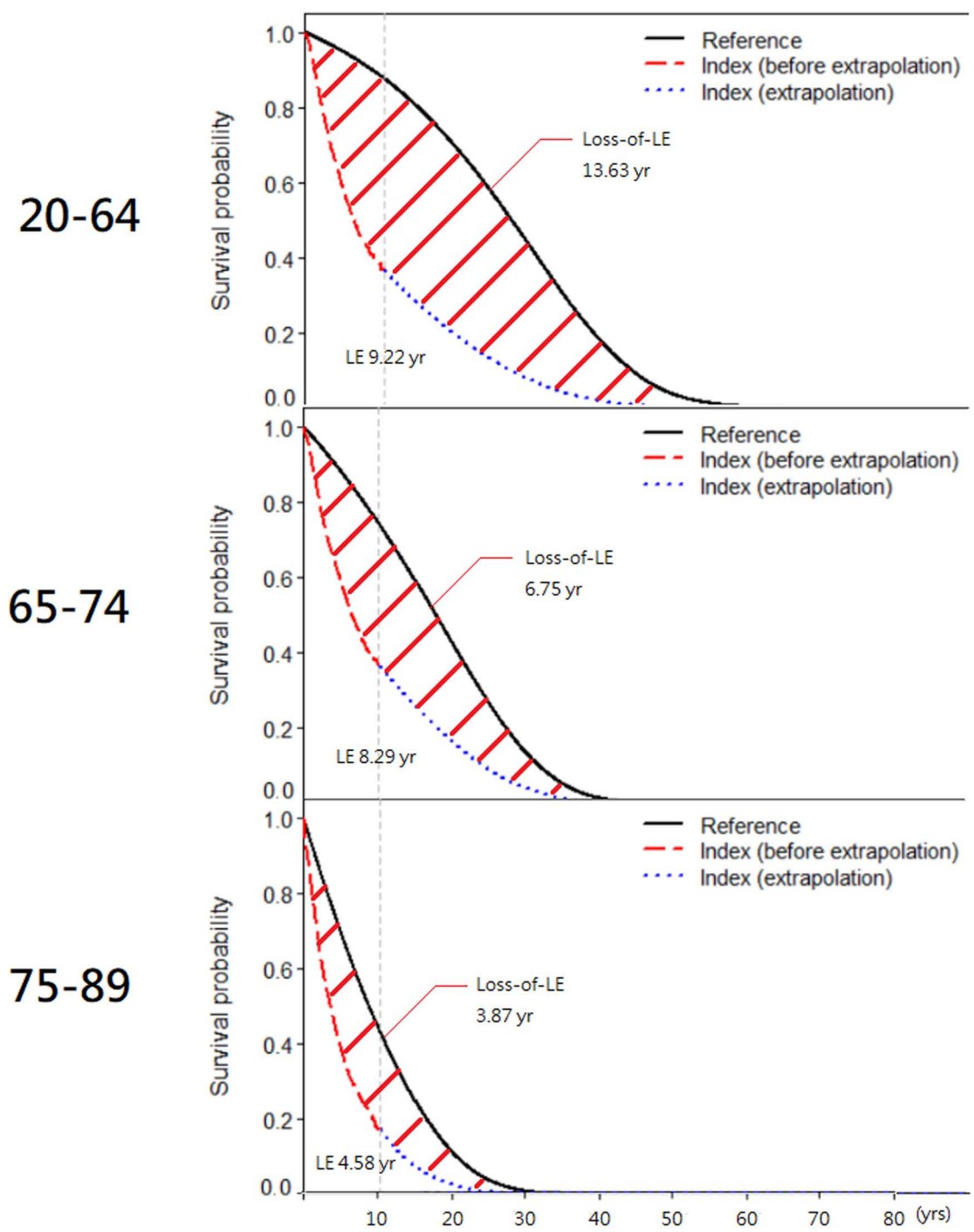

**Fig 1. Stage 4 Prostate cancer life expectancy and loss-of-life expectancy stratified by three age groups at diagnosis.** Red lines represent data from the PCa cohort; blue dashed lines represent age-, calendar year-matched referents simulated from the national statistics of Taiwan. The area between the black solid line and red/blue dashed lines indicates loss-of-life expectancy.

rate of synchronous metastatic PCa when compared to the United States. (32% vs. 5%) [17]. Based on the data presented, it's evident that a significant number of Taiwanese PCa patients receive late-stage diagnoses. Early detection of PCa has the potential to substantially decrease the incidence of synchronous metastatic PCa in Taiwan.

**Table 2. Estimation of life expectancy, loss of life expectancy, lifetimes cost (USD) and means cost per year for prostate cancer, stratified by age and prostatic specific antigen (PSA).**

| Age group | PSA (ng/dL) | N | No. of death | Age* | LE (SE)# | Loss of LE (SE) | Lifetime cost (SE) | Cost per year$ |
|---|---|---|---|---|---|---|---|---|
| 20–64 | | | | | | | | |
| | < 4 | 686 | 115 (16.76%) | 60.38 ± 3.78 | 18.55 (2.16) | 4.03 (2.16) | 66,098 (5,272) | 4,139 |
| | 4-9 | 1,899 | 66 (3.49%) | 59.98 ± 4.00 | – | – | – | – |
| | 10-19 | 1,309 | 89 (6.8%) | 60.38 ± 3.84 | – | – | – | – |
| | 20-50 | 881 | 113 (12.83%) | 60.59 ± 3.79 | 17.85 (3.02) | 4.49 (3.03) | 82,336 (6,781) | 5,250 |
| | >50 | 1,435 | 575 (40.07%) | 60.18 ± 4.03 | 9.16 (1.34) | 13.51 (1.36) | 82,887 (6,638) | 9,648 |
| 65–74 | | | | | | | | |
| | < 4 | 1,280 | 301 (23.52%) | 70.29 ± 2.85 | 12.86 (1.06) | 2.31 (1.05) | 56,599 (3,215) | 4,771 |
| | 4-9 | 2,735 | 244 (8.92%) | 69.92 ± 2.87 | – | – | – | – |
| | 10-19 | 2,384 | 261 (10.95%) | 70.10 ± 2.94 | – | – | – | – |
| | 20-50 | 1,909 | 342 (17.92%) | 70.50 ± 2.83 | 13.38 (1.29) | 1.68 (1.29) | 70,688 (4,082) | 5,684 |
| | >50 | 2,907 | 1,205 (41.45%) | 70.49 ± 2.87 | 8.87 (0.57) | 6.18 (0.57) | 69,031 (2,967) | 8,237 |
| 75–89 | | | | | | | | |
| | < 4 | 1,500 | 661 (44.07%) | 81.26 ± 4.07 | 7.18 (0.37) | 1.47 (0.38) | 32,892 (1,207) | 4,709 |
| | 4-9 | 1,700 | 479 (28.18%) | 79.93 ± 3.71 | – | – | – | – |
| | 10-19 | 2,214 | 697 (31.48%) | 80.28 ± 3.73 | – | – | – | – |
| | 20-50 | 2,395 | 964 (40.25%) | 80.75 ± 3.81 | 7.53 (0.34) | 1.33 (0.34) | 43,763 (1,624) | 5,938 |
| | >50 | 4,808 | 2,918 (60.69%) | 81.45 ± 3.95 | 4.74 (0.16) | 3.78 (0.16) | 36,005 (833) | 7,657 |

*Age presented as mean ± standard deviation;

#SE, standard error of the mean;

$1 dollar = $30.93 new Taiwan dollar; LE, Life expectancy, in year; loss of LE, loss of life expectancy.

Health care expenditures are crucial for shaping national health policy and allocating public health resources within a finite budget. Early detection of PCa through PSA screening and digital rectal examination may allow for early disease stratification, thereby improving disease prognosis and enabling treatment before disease progression. Our data (Table 1) support this point of view: there is no significant difference in the statistics of LE, loss of LE, and lifetime health care costs between stages 1-3 of PCa. In end-stage PCa, it causes a loss of LE by 13.63, 6.75, and 3.87 years in the 20-64, 65-74, and 75-89 age groups, respectively. The lifetime cost of end-stage PCa is $84,089, $68,109 and $35,841 in the 20-64, 65-74, and 75-89 age groups, respectively. In Table 2, the annual cost of end-stage PCa stratified by PSA level is observed across all three age groups ($9,648, $8,237, and $7,657, respectively), all of which are higher than the average annual cost of PCa in Taiwan ($3,818) according to Taiwan National Health Insurance database [18]. Early detection of PCa can save patients from experiencing a loss of LE and reduce the total lifetime cost and national health care expenditures.

PCa is the fifth leading cause of cancer-related deaths in Taiwan. However, routine PSA screening is not standard in Taiwan. There remains controversy of PSA screening for PCa within countries. A median follow-up of 15 years study enrolled with 415,357 male proved that a single invitation for PSA screening reduced PCa death [19]. However, systematic review and meta-analysis concluded that no estimated lifetime gained with PSA testing screening [20]. Currently, only the five major cancers in Taiwan, namely colorectal cancer, oral cancer, cervical cancer, lung cancer, and breast cancer, have screening plans subsidized by the government. Taiwan's national health system disapproves of population-wide PSA screening programs for PCa. The lower incidence and mortality rate of PCa in Taiwan compared to those in Europe and the United States is a major reason for this decision. The health-care

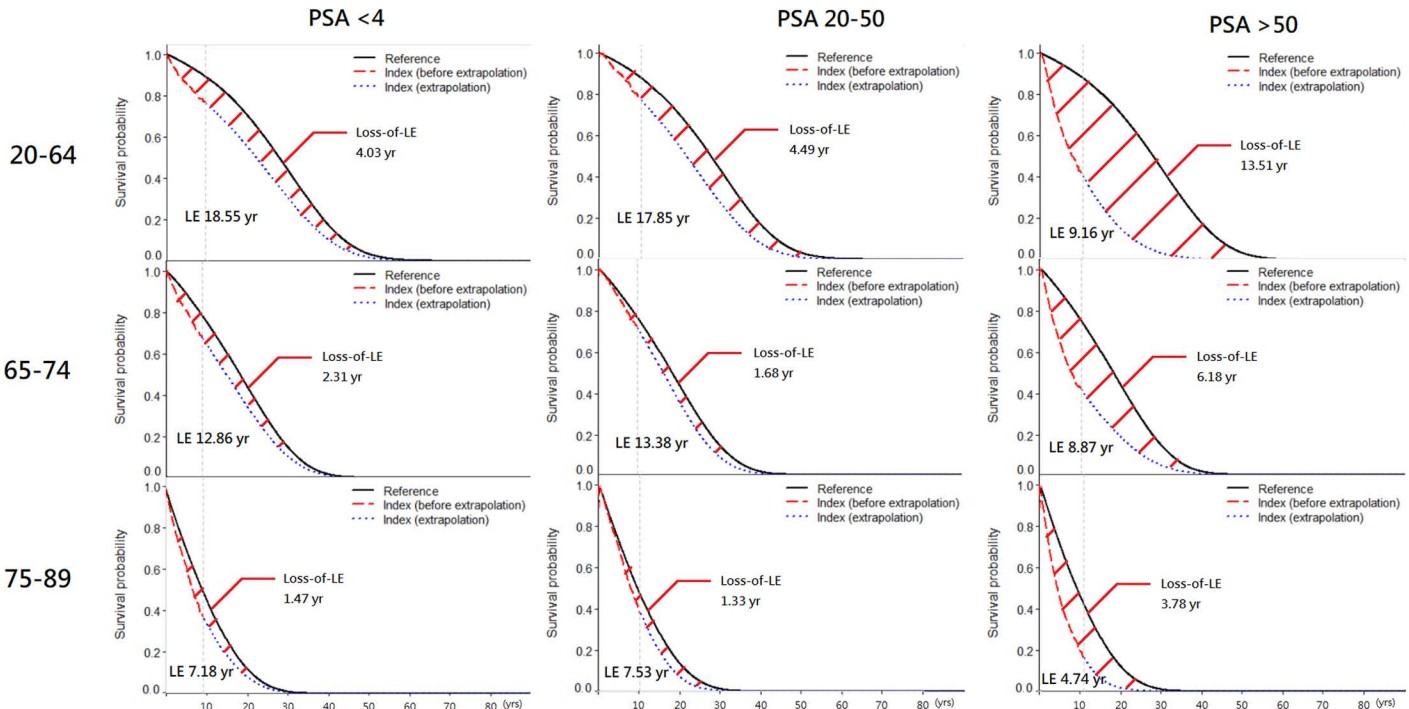

**Fig 2. Prostate cancer life expectancy and expected loss-of-life expectancy stratified by three age groups and PSA level at diagnosis.**

authority considers population-wide PSA screening to have low cost-effectiveness. Apart from this, the government has ignored the characteristics of PCa in Taiwan: a high proportion of synchronous metastatic PCa compared with other developed countries. On the other hand, unnecessary PSA screening might cause overdiagnosis and overtreatment, doing more harm than benefits to the patients [16]. The Taiwan Urological Association recommends early PSA testing only in male aged more than 50 year-old or male more than 45 year-old with family history of PCa. However, early PCa has few symptoms and is often diagnosed as advanced PCa while symptomatic. As a result, many men diagnosed with PCa never know they have the disease unless they undergo an examination. To make matters worse, Taiwan's national health care system does not cover approaches like Magnetic Resonance Imaging and biomarkers, such as the Prostate Health Index, for the diagnosis of PCa.

Taiwan has had a national health insurance system since 1995, covering most of the medical expenses for PCa patients. This coverage extends from treatment modalities of radical prostatectomy, internal or external radiation therapy in the localized cancer, to androgen deprivation therapy, second-generation antiandrogens, and chemotherapy in the advanced stage of the disease. However, the nature of social welfare, universal cost coverage of national health insurance also places a heavy economic burden on itself. The increasing incidence and mortality rate of PCa make it a critical public health issue, posing a great economic burden on the national health system. Despite the growing number of therapies and treatments for PCa, it is associated with substantial societal costs and healthcare expenses. These costs are expected to rise with the incorporation of novel and more expensive technologies for managing PCa in the future. Early detection and treatment of PCa can contribute to reducing overall PCa medical costs. This is particularly relevant for the considerable medical costs associated with synchronous metastatic PCa. If the national healthcare system were to initiate strategies such as the introduction of a PSA screening program, it could further decrease overall PCa

**Table 3. Estimation of life expectancy, loss of life expectancy, lifetimes cost (USD) and means cost per year for prostate cancer, stratified by age and Gleason's score.**

| Age group | Gleason's score | N | No. of death | Age* | LE (SE)# | Loss of LE (SE) | Lifetime cost (SE) | Cost per year$ |
|---|---|---|---|---|---|---|---|---|
| 20–64 | | | | | | | | |
| | ≤ 6 | 2,118 | 102 (4.82%) | 60.25 ± 3.87 | | | | |
| | 3 + 4 | 999 | 63 (6.31%) | 60.26 ± 3.94 | | | | |
| | 4 + 3 | 816 | 96 (11.76%) | 60.25 ± 3.92 | 16.91 (3.4) | 5.75 (3.42) | 77,071 (7,569) | 5,068 |
| | 8 | 827 | 143 (17.29%) | 60.3 ± 3.82 | 16.75 (2.15) | 5.84 (2.15) | 91,875 (6,472) | 6,234 |
| | 9-10 | 1,151 | 454 (39.44%) | 60.24 ± 4.03 | 10.1 (1.48) | 12.52 (1.47) | 79,800 (6,667) | 8,565 |
| 65–74 | | | | | | | | |
| | ≤ 6 | 3,346 | 348 (10.4%) | 70.04 ± 2.89 | | | | |
| | 3 + 4 | 1,849 | 250 (13.52%) | 70.19 ± 2.88 | – | – | – | – |
| | 4 + 3 | 1,700 | 294 (17.29%) | 70.39 ± 2.87 | 13.3 (1.56) | 1.8 (1.56) | 66,139 (4,099) | 5,335 |
| | 8 | 1,602 | 374 (23.35%) | 70.3 ± 2.83 | 11.63 (1.35) | 3.51 (1.35) | 73,643 (4,930) | 6,748 |
| | 9-10 | 2,316 | 902 (38.95%) | 70.48 ± 2.9 | 10.01 (0.82) | 5.03 (0.83) | 70,173 (3,750) | 7,548 |
| 75–89 | | | | | | | | |
| | ≤ 6 | 2,693 | 818 (30.38%) | 80.32 ± 3.81 | | | | |
| | 3 + 4 | 1,734 | 615 (35.47%) | 80.39 ± 3.74 | – | – | – | – |
| | 4 + 3 | 1,997 | 805 (40.31%) | 80.67 ± 3.8 | 7.23 (0.47) | 1.69 (0.47) | 42,540 (1,857) | 5,984 |
| | 8 | 2,045 | 912 (44.6%) | 80.98 ± 3.96 | 7.2 (0.35) | 1.56 (0.36) | 43,089 (1,802) | 6,152 |
| | 9-10 | 3,468 | 2,085 (60.12%) | 81.43 ± 3.92 | 5.2 (0.22) | 3.32 (0.22) | 38,175 (1,159) | 7,483 |

*Age presented as mean±standard deviation;

#SE, standard error of the mean;

$1 dollar = $30.93 new Taiwan dollar; LE, Life expectancy, in year; loss of LE, loss of life expectancy.

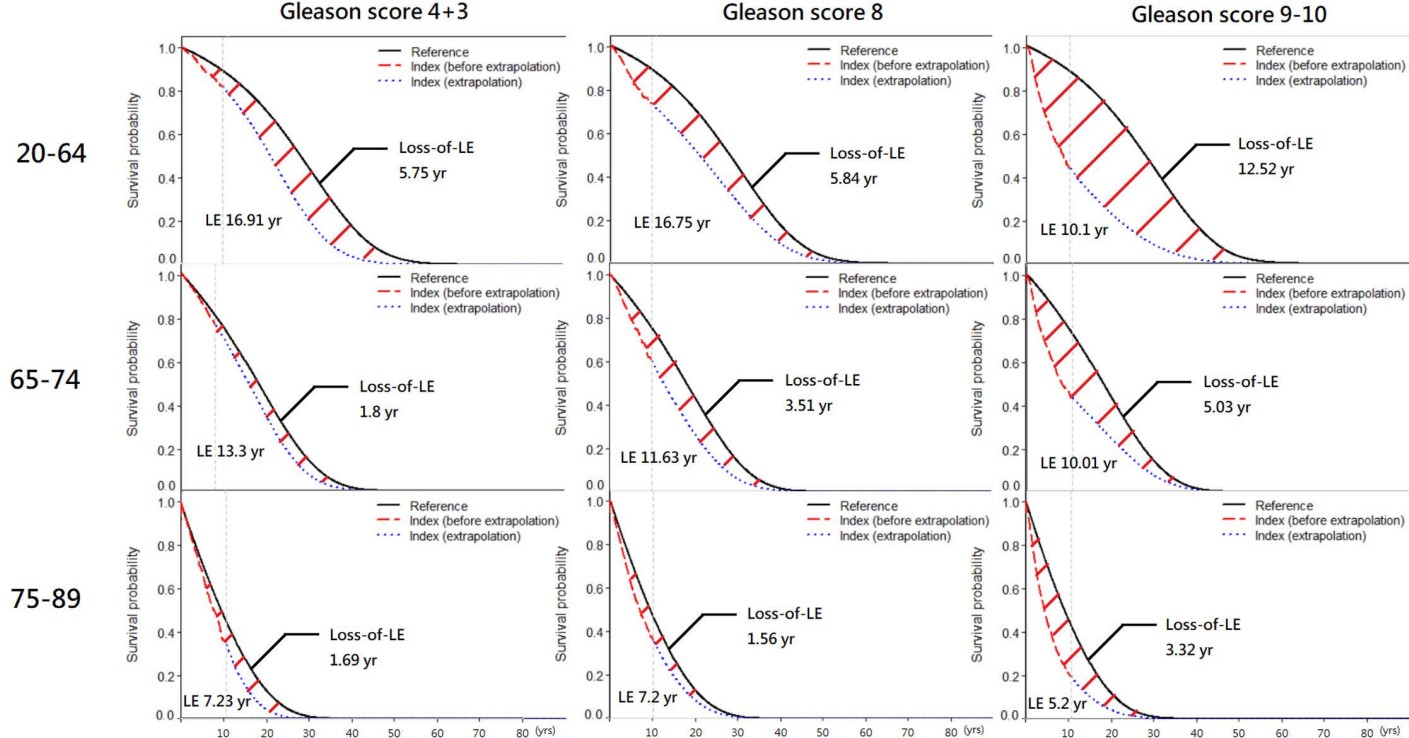

**Fig 3. Prostate cancer life expectancy and expected loss-of-life expectancy stratified by three age groups and Gleason score at diagnosis.**

costs and healthcare expenditures. This approach becomes especially crucial in Taiwan, where a high rate of synchronous metastatic PCa has been recorded.

The strength of our study lies in its nationwide cohort and long-term follow-up period. The data were obtained from the TCR database, which provides exhaustive, complete and high quality information on cancer medical records in Taiwan. Although, the limitation is that the study is inherently retrospective. Second of all, the total cost cannot exclude the cost caused by other diseases.

In conclusion, this study provides real-world evidence to support early detection of PCa can save LE and decrease the total cost burden on the healthcare system. The results of this study contribute to better national healthcare resource allocation and policy decisions in PCa treatment. Furthermore, it also raises awareness and public concern about PCa and highlights its impact on the people of Taiwan, both in terms of loss of LE and lifetime cost.

## Conclusions

This study provides real-world evidence to support health care policy-making and clinical decisions regarding PCa. Due to the high proportion of synchronous metastatic PCa in Taiwan, the findings of this analysis emphasize the importance of early detection of PCa, which can save LE and decrease the total cost burden on the healthcare system.

## Supporting information

**S1 Fig.  The proportion of PCa patients who underwent TURP one year before PCa diagnosis.**
(TIF)

## Acknowledgments

Collection and assembly of data: Ya-Chu Yang

## Author contributions

**Conceptualization:** Yi-Sheng Lin, Yen-Chuan Ou, Ying-Ming Chiu.

**Data curation:** Yi-Sheng Lin, Ying-Ming Chiu.

**Formal analysis:** Pin-Chun Liu.

**Methodology:** Yen-Chuan Ou, Ying-Ming Chiu.

**Project administration:** Yi-Sheng Lin, Ying-Ming Chiu.

**Supervision:** Yi-Sheng Lin, Yen-Chuan Ou, Ying-Ming Chiu.

**Validation:** Yen-Chuan Ou, Chao-Yu Hsu, Min-Che Tung.

**Visualization:** Pin-Chun Liu.

**Writing – original draft:** Pin-Chun Liu.

**Writing – review & editing:** Pin-Chun Liu, Yi-Sheng Lin.

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
