## [Decision Letter · Decision Letter 0]

7 Jan 2025

PONE-D-24-37580Comprehensive analysis of prostate cancer life expectancy, loss of life expectancy, and healthcare expenditures: Taiwan national cohort study spanning 2008 to 2019PLOS ONE

Dear Dr. Ou,

Thank you for submitting your manuscript to PLOS ONE. After careful consideration, we feel that it has merit but does not fully meet PLOS ONE’s publication criteria as it currently stands. Therefore, we invite you to submit a revised version of the manuscript that addresses the points raised during the review process.

We look forward to receiving your revised manuscript.

Kind regards,

Po-Fan Hsieh

Academic Editor

PLOS ONE

Journal Requirements:

2. Please note that your Data Availability Statement is currently missing [the repository name and/or the DOI/accession number of each dataset OR a direct link to access each database]. If your manuscript is accepted for publication, you will be asked to provide these details on a very short timeline. We therefore suggest that you provide this information now, though we will not hold up the peer review process if you are unable.

Reviewers' comments:

Reviewer's Responses to Questions

**Comments to the Author**

1. Is the manuscript technically sound, and do the data support the conclusions?

Reviewer #1: Yes

Reviewer #2: Yes

2. Has the statistical analysis been performed appropriately and rigorously? 

Reviewer #1: I Don't Know

Reviewer #2: Yes

3. Have the authors made all data underlying the findings in their manuscript fully available?

Reviewer #1: Yes

Reviewer #2: Yes

4. Is the manuscript presented in an intelligible fashion and written in standard English?

Reviewer #1: Yes

Reviewer #2: Yes

5. Review Comments to the Author

Reviewer #1: The authors reported the real-world and long-term follow-up evidence of PCa life expectancy, loss of life expectancy, and healthcare expenditures in the Taiwan region, emphasizing that early detection of PCa may save life expectancy and decrease the total cost. This study contains potential value on clinical applications and solving public health problems for Taiwan residents. However, several minor issues need to be addressed before acceptance:

1. In lines 51-52 the authors mentioned “the healthcare cost of synchronous metastatic PCa in all age groups is higher than the average cost for PCa patients”. But I couldn’t find the data that indicate the average cost for non-synchronous metastatic PCa in the tables. Please present this part of data in an appropriate way.

2. I recommend the authors draw figures to better illustrate the noteworthy variations in loss of life expectancy, as previous similar studies (PMID: 37152349) (PMID: 33295139) showed.

3. More recent references are required in the manuscript.

Reviewer #2: Dear Authors,

Thank you for submitting your manuscript. This study presents an insightful analysis of prostate cancer (PCa) in Taiwan, and I appreciate the opportunity to review it. Below are my comments and suggestions for improvement:

Abstract:

- In the statement, “The loss of LE in the three groups is 13.63, 6.75, and 3.87 years, respectively,” it is unclear whether this refers to the three risk categories or the three age groups. Please clarify.

- Regarding the sentence, “The healthcare cost of synchronous metastatic PCa in all age groups is higher than the average cost for PCa patients in Taiwan,” I suggest including some brief data to support this statement.

Materials and Methods:

- The staging of PCa is not mentioned. Did you use the D’Amico classification, the NCCN classification, or another system? Additionally, how did you group the patients into the four-stage categories?

Thank you again for your valuable work and for allowing me to review your manuscript.

6. PLOS authors have the option to publish the peer review history of their article (what does this mean? ). If published, this will include your full peer review and any attached files.

**Do you want your identity to be public for this peer review?** For information about this choice, including consent withdrawal, please see our Privacy Policy .

Reviewer #1: No

Reviewer #2: **Yes: ** Bignante Gabriele, MD

---

## [Author Response · Author response to Decision Letter 1]

12 Feb 2025

Response to Reviewer #1

Q1: 1. In lines 51-52 the authors mentioned “the healthcare cost of synchronous metastatic PCa in all age groups is higher than the average cost for PCa patients”. But I couldn’t find the data that indicate the average cost for non-synchronous metastatic PCa in the tables. Please present this part of data in an appropriate way.

A1: The annually average cost of prostate cancer was added in line 255.

Q2: 2. I recommend the authors draw figures to better illustrate the noteworthy variations in loss of life expectancy, as previous similar studies (PMID: 37152349) (PMID: 33295139) showed.

A2: The illustrations were added and labeled as figure 1-3, stratified by age PSA level and Gleason score respectively.

Q3: 3. More recent references are required in the manuscript.

A3: Several new references were cited and added at discussion section. (References 19-20)

Responding to Reviewer #2

Q1: In the statement, “The loss of LE in the three groups is 13.63, 6.75, and 3.87 years, respectively,” it is unclear whether this refers to the three risk categories or the three age groups. Please clarify.

A1: It is refers as three age groups: ages 20-64 (13.63 years); ages 65-74 (6.75 years) and ages 75-89 (3.87 years)

Q2: Regarding the sentence, “The healthcare cost of synchronous metastatic PCa in all age groups is higher than the average cost for PCa patients in Taiwan,” I suggest including some brief data to support this statement.

A2: The annually average cost of prostate cancer was added in line 255.

Q3: The staging of PCa is not mentioned. Did you use the D’Amico classification, the NCCN classification, or another system? Additionally, how did you group the patients into the four-stage categories?

A3: This study uses the AJCC cancer staging (8th edition) for staging prostate cancer. The data was collected from Taiwan Cancer Registry database which included prostate cancer stage. The first reason for using four-stage categories is that it is the same as the staging system currently used clinically. Secondly, it facilitates communication and discussion among research members.

---

## [Editor Report · Decision Letter 1]

17 Feb 2025

Comprehensive analysis of prostate cancer life expectancy, loss of life expectancy, and healthcare expenditures: Taiwan national cohort study spanning 2008 to 2019

PONE-D-24-37580R1

Dear Dr. Ou,

We’re pleased to inform you that your manuscript has been judged scientifically suitable for publication and will be formally accepted for publication once it meets all outstanding technical requirements.

Kind regards,

Po-Fan Hsieh

Academic Editor

PLOS ONE

---

## [Editor Report · Acceptance letter]

PONE-D-24-37580R1

PLOS ONE

Dear Dr. Ou,

I'm pleased to inform you that your manuscript has been deemed suitable for publication in PLOS ONE. Congratulations! Your manuscript is now being handed over to our production team.

Kind regards,

on behalf of

Dr. Po-Fan Hsieh

Academic Editor

PLOS ONE